# *Snap29* mutant mice recapitulate neurological and ophthalmological abnormalities associated with 22q11 and CEDNIK syndrome

Vafa Keser[1], Jean-François Boisclair Lachance [1], Sabrina Shameen Alam[1], Youngshin Lim [2], Eleonora Scarlata[3,4], Apinder Kaur[1], Tian Fang Zhang[5], Shasha Lv[5], Pierre Lachapelle[5], Cristian O'Flaherty[3,4], Jeffrey A. Golden[2] & Loydie A. Jerome-Majewska[1,6,7]*

Synaptosomal-associated protein 29 (*SNAP29*) encodes a member of the SNARE family of proteins implicated in numerous intracellular protein trafficking pathways. *SNAP29* maps to the 22q11.2 region and is deleted in 90% of patients with 22q11.2 deletion syndrome (22q11.2DS). Moreover, bi-allelic *SNAP29* mutations in patients are responsible for CEDNIK (cerebral dysgenesis, neuropathy, ichthyosis, and keratoderma) syndrome. A mouse model that recapitulates abnormalities found in these syndromes is essential for uncovering the cellular basis of these disorders. In this study, we report that mice with a loss of function mutation of *Snap29* on a mixed CD1;FvB genetic background recapitulate skin abnormalities associated with CEDNIK, and also phenocopy neurological and ophthalmological abnormalities found in CEDNIK and a subset of 22q11.2DS patients. Our work also reveals an unanticipated requirement for *Snap29* in male fertility and supports contribution of hemizygosity for *SNAP29* to the phenotypic spectrum of abnormalities found in 22q11.2DS patients.

[1] Department of Human Genetics, McGill University, Montreal, QC H4A 3J1, Canada. [2] Department of Pathology, Brigham and Women's Hospital, Harvard Medical School, Boston, MA 02115, USA. [3] Department of Pharmacology and Therapeutics, McGill University, Montreal, QC H4A 3J1, Canada. [4] Department of Surgery (Urology Division), McGill University, Montreal, QC H4A 3J1, Canada. [5] Department of Ophthalmology & Visual Sciences, McGill University Health Centre at Glen Site, Montreal, QC H4A 3J1, Canada. [6] Department of Anatomy and Cell Biology, McGill University Health Centre at Glen Site, Montreal, QC H4A 3J1, Canada. [7] Department of Pediatrics, McGill University, Montreal, QC H4A 3J1, Canada. *email: loydie.majewska@mcgill.ca

**22**q11.2 deletion syndrome (22q11.2DS), also known as DiGeorge and Velocardial Facial syndrome, is the most common contiguous gene syndrome found in newborns—at a frequency of 1:3000–1:6000[1]. The deletion, which is mediated by low copy repeats on 22q11.2, results in de novo deletion of 3.0 MB of chromosome 22q11.2 in 90% of patients, and haploinsufficiency of over 40 protein coding genes[2]. 22q11.2DS patients have heterogeneous clinical presentations associated with multi-organ dysfunction including cardiac and palate abnormalities, hypoparathyroidism, neurological and motor defects, epilepsy, cognitive deficits, and neuropsychiatric illness (e.g., schizophrenia). These dysfunctions show incomplete penetrance and variable expressivity depending on the patient's age[1].

CEDNIK (cerebral dysgenesis, neuropathy, ichthyosis, and keratoderma) patients with intact chromosome 22q11.2 carry homozygous mutations in *SNAP29*. These patients present with a number of clinical manifestations, some of which overlap with those found in 22q11.2DS. CEDNIK patients exhibit skin defects at birth or in the first few months of life, failure to thrive, cerebral malformations, developmental delay, epilepsy, severe mental retardation, roving eye movements during infancy, trunk hypotonia, poor head control, craniofacial dysmorphisms, and mild deafness[3–5].

In humans, *SNAP29* mutations show variable expressivity and incomplete penetrance. For example, patients homozygous for the c. 486_487insA mutation may present with the full constellation of CEDNIK or only with motor and neurological abnormalities (polymicrogyria, trunk hypotonia, dysplastic or absent corpus callosum, epilepsy, and hypoplastic optic nerves) but no dermatological abnormalities or dysmorphic features[3,6]. Similarly, we reported that only one of four patients with hemizygous deletion of 22q11.2 and deleterious variants in the remaining *SNAP29* allele showed both ichthyosis and neurological manifestations[7]. Whereas, two of the four patients were atypical and did not have any skin abnormalities. Finally, though one patient with ADNFLE (autosomal dominant nocturnal frontal lobe epilepsy) was shown to carry a single heterozygous truncating mutation in *SNAP29*[8], abnormalities have not been reported in heterozygous parents of children with CEDNIK syndrome, consistent with the hypothesis that mutations in *SNAP29* result in variable expressivity and incomplete penetrance.

To date, one mutant mouse model with deletion of exon 2 of *Snap29* has been generated and characterized. This mutant mouse line was made on the inbred C57BL/6 genetic background. Although *Snap29* homozygous mutant pups on this genetic background model skin abnormalities found in CEDNIK patients, they die at birth[9]. Since penetrance and expressivity of mutations in mouse models are modified by genetic background, we generated mice with deletion of exon 2 of *Snap29* on a mixed CD1; FvB genetic background. In this study, we show that mice with constitutive loss of function mutation in *Snap29* on this mixed genetic background survive and exhibit: skin abnormalities, neurological defects, facial dysmorphism, psychomotor retardation, and fertility problems with variable expressivity and incomplete penetrance. Our data indicate that abnormalities associated with mutations in *SNAP29* in human patients with 22q11.2DS and CEDNIK can be recapitulated in mice. This mouse model can now be used to uncover the etiology of abnormalities found in patients with mutations of *SNAP29*.

## Results

### *Snap29* is ubiquitously expressed during mouse embryogenesis.

We first examined the expression pattern of *Snap29* from E9.5 to E12.5 using digoxigenin-labeled RNA probes and in situ hybridization. From E9.5 onward, ubiquitous expression of *Snap29* was found by wholemount (Supplementary Fig. 1) and section (Fig. 1a) in situ hybridization. Thus, *Snap29* is expressed in derivatives of all three germ layers.

### Generation of *Snap29* mutant mouse line on a mixed CD1; FvB genetic background.

To generate a mutant mouse line with a loss of function mutation in *Snap29* on a mixed genetic background we used CRISPR/Cas9 to delete exon 2 of *Snap29* and create a frameshift and premature termination signal[9]. Of 14 mice born, four females carried Sanger-sequence verified deletions of 464 and 517 bp, in two mutant mouse lines *Snap29*[lam1] and *Snap29*[lam2], respectively. Male offspring carrying these deletions, which included exon 2 as well as 267 and 320 bp of flanking intronic sequences, respectively (Fig. 1b) were used to establish a *Snap29* mutant colony.

### *Snap29* homozygous mutants (*Snap29*[−/−]) on the CD1; FvB genetic background survive to adulthood.

*Snap29* heterozygous (*Snap29*[+/−]) mice showed no apparent morphological abnormalities, survived to adulthood, and were fertile. *Inter se* mating of *Snap29*[+/−] mice revealed normal Mendelian segregation of *Snap29* mutant alleles at birth (Table 1). Furthermore, *Snap29*[−/−] mice had no detectable protein (Fig. 1c, d, Supplementary Fig. 2) and survived to adulthood (n = 27 of 40) (Table 1). Thus, we concluded that, loss of function mutations of *Snap29* on the CD1; FvB mixed genetic background is compatible with survival to adulthood.

### *Snap29*[−/−] mutants recapitulate variable expressivity and penetrance of skin abnormalities found in CEDNIK.

To assess if *Snap29*[−/−] mutant pups and adult mice exhibited pathologies and abnormalities found in 22q11.2DS and CEDNIK patients, we followed 40 *Snap29*[−/−] mutant pups, from 18 litters, from birth until weaning. A small fraction of these homozygous mutant pups, with no apparent defects, died within 2 days of birth (n = 2 of 40). However, the majority of homozygous mutant pups survived the perinatal period and developed skin defects between P2 and P6 (n = 33). These abnormalities were classified as severe, moderate, or mild depending on the extent of scaling or peeling observed (Fig. 2 and Supplementary Table 1). Furthermore, although some of the 33 homozygous mutant pups that developed skin abnormalities died between P3 and P7 (n = 12) most survived (n = 21). In addition, surviving homozygous mutant pups recovered from their skin defects and formed fur coats that were undistinguishable from their litter mates. Nonetheless, by weaning, 100% of surviving homozygous mutants had thickened and reddish ears, scaling on their ears and paws (Fig. 2e, g), and swollen reddish genitalia (Fig. 2i), including those with no obvious skin defects in the early perinatal period (n = 5). Thus, *Snap29*[−/−] mutant mice on a mixed genetic background recapitulate variable expressivity of skin abnormalities found in CEDNIK patient.

To determine the basis of skin defects in *Snap29*[−/−] mutant pups, histological analysis of dorsal skin was performed. Hematoxylin and Eosin (H&E) staining at E16.5 revealed no

**Table 1 Number of embryos or pups per *Snap29* genotype**

| Stage | +/+ (dead) | +/- (dead) | -/- (dead) | # litters |
|---|---|---|---|---|
| E11.5 | 28 (0) | 45 (0) | 29 (0) | 9 |
| E16.5 | 36 (0) | 51 (0) | 33 (0) | 10 |
| P1 | 37 (0) | 52 (0) | 39 (0) | 12 |
| P7 | 33 (0) | 84 (0) | 40 (13)* | 17 |

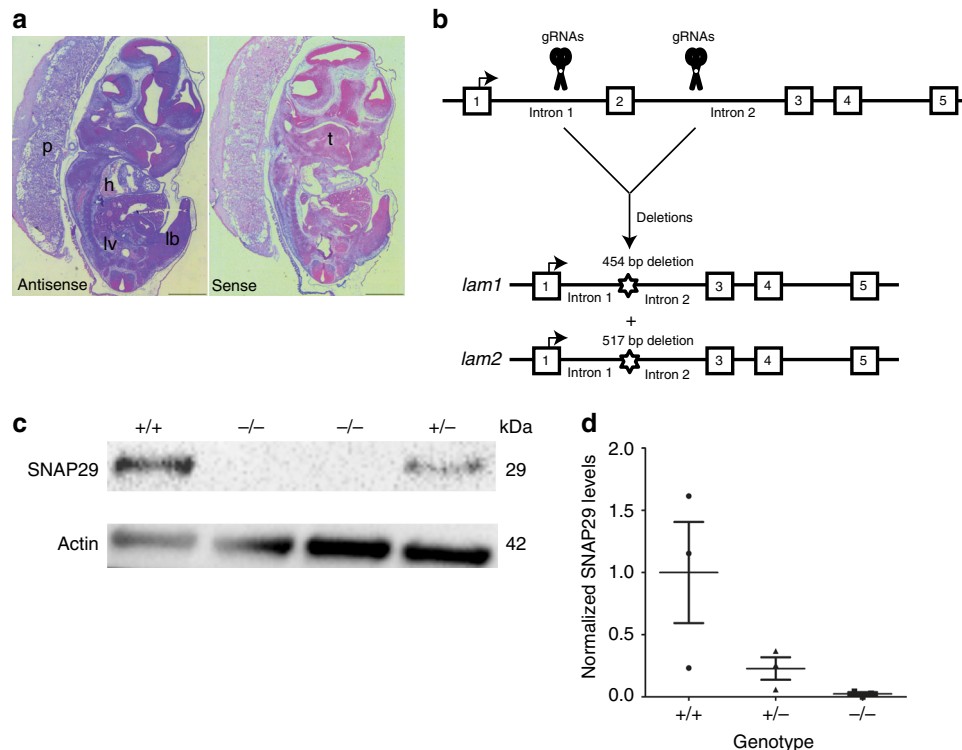

**Fig. 1** *Snap29* mRNA is ubiquitously expressed and CRISPR-mediated targeting of exon 2 depletes SNAP29 protein. **a** Antisense (left) and sense (right) probe of *Snap29* mRNA expression at E12.5. **b** CRISPR design and the resulting targeted alleles, one with a 454 bp and one with a 517 bp deletion. **c** Western blot analysis of skin preps of mice carrying either wild-type, heterozygous, or homozygous mutant for the *Snap29* allele. **d** Quantification of SNAP29 expression. SNAP29 levels are normalized to WT. Lb limbud, fore, or hind, l lung, hrt heart, s somite, tb tailbud. Error bar represent: standard deviation. Scale bar equals 1000 μm

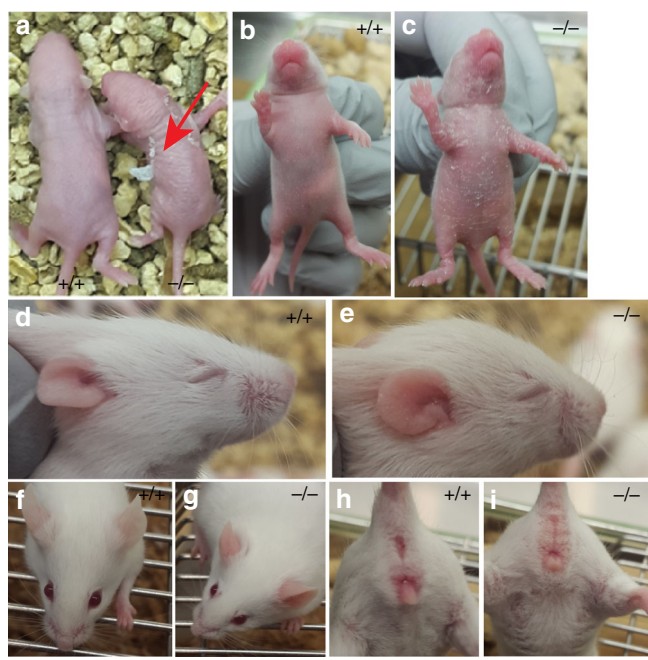

**Fig. 2** *Snap29* mutant mice show diverse skin defects. **a** Example of the severe skin peeling observed in *Snap29*−/− P2 pup (red arrow). **b**, **c** P3 pups showing mild skin peeling. **d**–**e** P11 pups abnormal head shape and ichtyosis on ear. **f**, **g** Example of reddish and thickened ears at P30. **h**–**i** Example of swollen reddish genitalia

morphological differences between wild-type, *Snap29*+/−, and homozygous mutant samples (Supplementary Fig. 3A–C). However, at P1—prior to onset of skin abnormalities and P3—when skin abnormalities were morphologically apparent, epidermis of *Snap29*−/− mutant pups showed hyperkeratosis and condensed stratum corneum (Fig. 3a–c). In addition, transmission electron microscopy of P1 dorsal skin (Fig. 3d, e) also revealed remnants of organelles in the lower layers of the stratum granulosum (Fig. 3f, g) similar to those previously reported on the C57Bl/6 genetic background[9]. Thus, the epidermal structure is defective in pups with a loss of function mutation of *Snap29* on the CD1;FvB mixed genetic background.

**_Snap29_−/− mutants form a functional skin barrier**. We postulated that *Snap29*−/− mutants with epidermal skin defects survived the neonatal period because a functional skin barrier forms. To test this hypothesis, an X-Gal skin permeability assay was used to evaluate the integrity of the epidermal barrier of E17.5 and P1 litters. At E17.5, 30% of wild type ($n = 3$ of 10), 47% of *Snap29*+/− ($n = 8$ of 18), and 80% of *Snap29*−/− ($n = 12$ of 14) mutant embryos, showed X-Gal staining on the ventral surface of the body wall ($p = 0.034$) (Supplementary Fig. 4A–C), demonstrating a significant delay of skin barrier formation in *Snap29*−/− mutants (Supplementary Table 2). However, at P1 no X-Gal staining was found in wild type ($n = 14$), *Snap29*+/− ($n = 29$) or *Snap29*−/− mutant pups ($n = 11$) (Supplementary Fig. 4D–F). Thus, although skin barrier formation is delayed in homozygous mutant embryos, a proper skin barrier was present at birth.

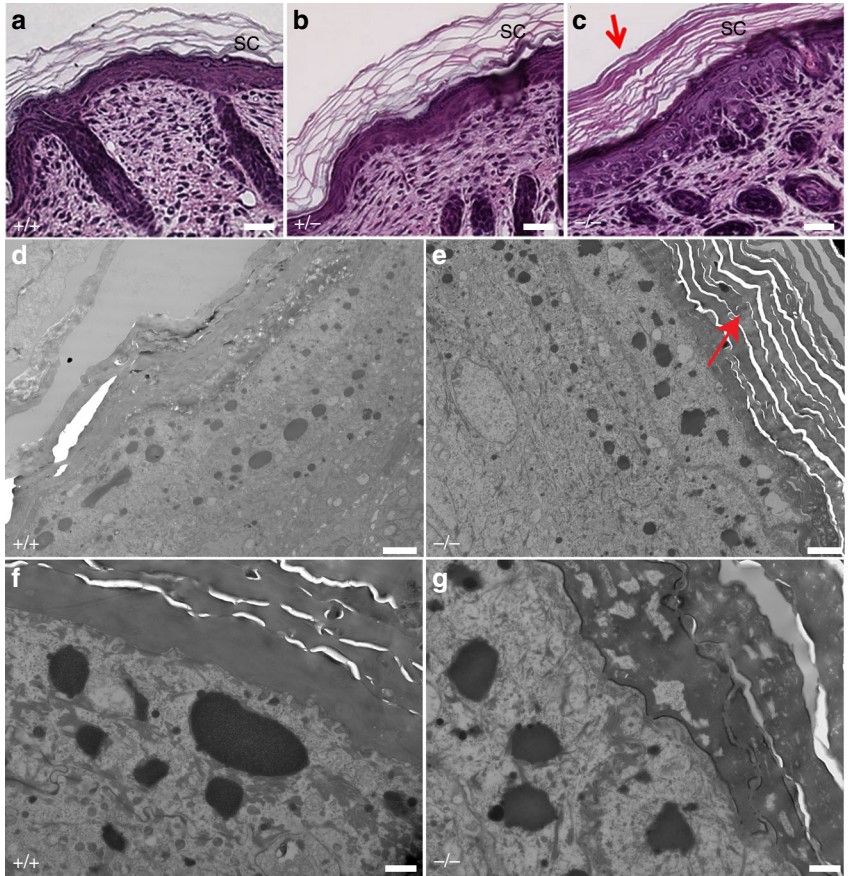

**Fig. 3** *Snap29* mutant mice show signs of hyperkeratosis. Hematoxylin and eosin staining (**a–c**) and TEM (**d–g**) of dorsal skin. The stratum corneum (SC) in *Snap29*$^{-/-}$ mice was found thicker and more condensed (**c**, **e**, red arrow) than that of the *Snap29*$^{+/+}$ or *Snap29*$^{+/-}$ mice. Abnormal structures were seen in the extracellular spaces between granulocytes and lower corneocytes. Scale bars represent 25 μm (**a–c**), 2 μm (**d**, **e**), and 500 nm (**f**, **g**)

***Snap29*$^{-/-}$ mutants display skeletal abnormalities with incomplete penetrance and variable expressivity.** 22q11.2DS and CEDNIK patients have skeletal abnormalities as well as dysmorphic facial features[4]. To determine if *Snap29*$^{-/-}$ mutant mice phenocopy any of these abnormalities, we performed Alcian blue and Alizarin red staining to visualize cartilage and bones of *Snap29*$^{-/-}$ mutants at four stages (E14.5, E16.5, P1, and P3). Skeletal abnormalities were found at low penetrance in the face, the head and ribs of *Snap29*$^{-/-}$ mutants at all stages. The most common defect found was abnormal mineralization. Mineralization of the parietal and occipital bones was either delayed ($n = 3$ of 14) or precocious ($n = 6$ of 14) in homozygous mutants (Fig. 4d–g) when compared with wild type ($n = 11$) and *Snap29*$^{+/-}$ ($n = 25$) litter mates. In addition, *Snap29*$^{+/-}$ ($n = 8$ of 17) and *Snap29*$^{-/-}$ mutants ($n = 13$ of 32) had extra lumbar ribs (14 pairs of ribs instead of 13) while only a single such case was seen in one wildtype embryo ($n = 12$) (Fig. 4a–c). Thus, *Snap29*$^{-/-}$ mutant mice on a mixed genetic background model a subset of skeletal abnormalities and dysmorphisms found in patients with mutations in *SNAP29*, though these phenotypes showed incomplete penetrance and variable expressivity.

***Snap29*$^{-/-}$ mutant mice have motor defects.** Newborn *Snap29*$^{-/-}$ mutants, in general, appeared to move slower than their wild-type and heterozygous litter mates. In addition, some were unable to stand up on their feet after turning onto their stomach ($n = 6$ of 11) when compared with wildtype ($n = 0$ of 13) and heterozygous ($n = 0$ of 21) littermates (Supplementary Movie 1, Supplementary Fig. 5A, B). To determine if motor defects persist in adult *Snap29*$^{-/-}$ mutants, a rotarod was used to evaluate neuromuscular coordination, balance and grip strength of male mice[10] at 5 weeks of age. We found that *Snap29* homozygous mutants ($n = 3$) had a shorter latency to fall when compared with wild type ($n = 3$) and heterozygous ($n = 3$) (Supplementary Fig. 6). These findings suggest that *Snap29* homozygous mutants exhibit deficiencies in neuromuscular coordination, balance, and/or grip strength.

***Snap29*$^{-/-}$ mutant mice move slower and show forelimb preferences.** We also assessed locomotion and coordination of 6-week old male and female mice using the CatWalk system[11]. We analyzed parameters measured using the CatWalk and grouped them into the 5 categories described by Gaballero-Garrido et al.[12]: Run characterization, Temporal, Spatial Kinetic and Interlimb coordination. We found that *Snap29*$^{-/-}$ mutant male ($n = 9$) and female ($n = 14$) mice showed deficiencies in all 5 categories when compared with sex and age matched litter mates (Figs. 5 and 6 and Supplementary Results and Figs. 7, 8). In addition, heterozygous females exhibited defects in interlimb coordination (Supplementary Fig. 7C, Supplementary Table 3). Parameters with differences in *Snap29*$^{-/-}$ mutant female and male mice indicate that they move slower than their littermate controls and show preferential use of their forelimbs for movement.

***Snap29*$^{-/-}$ mutants phenocopy hypotonia found in CEDNIK.** To test if deficiencies uncovered by the rotarod and Catwalk tests were due to changes in strength, a grip meter was used to measure strength of 7-week old male and female mice.

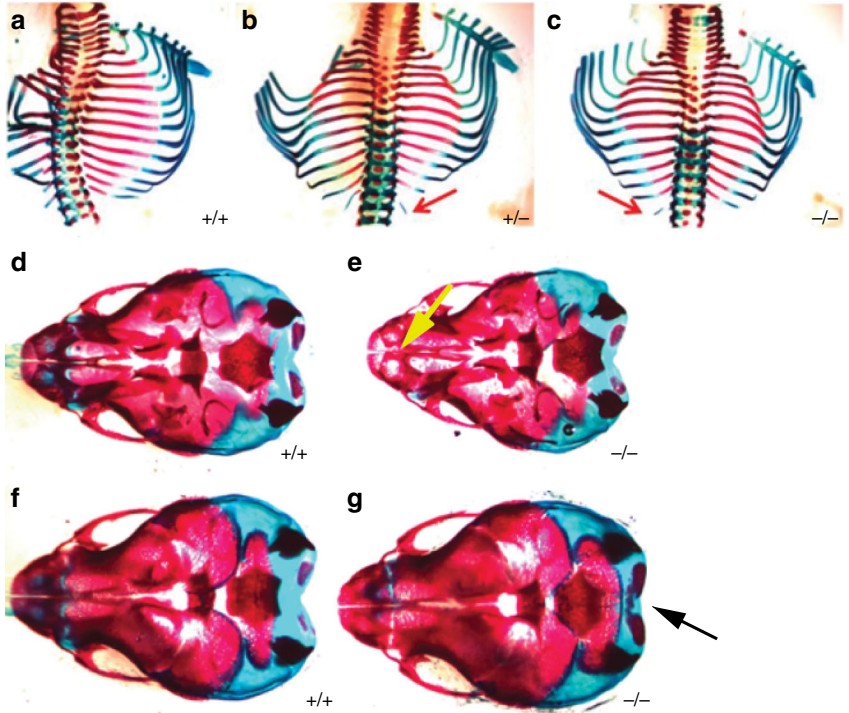

**Fig. 4** *Snap29* heterozygous and homozygous mutant mice have skeletal defects. Heterozygous (**b**) and homozygous mutant (**c**) embryos had extra lumbar ribs (red arrows) that were not seen in control mice (**a**). Advanced (**d**, **e**, yellow arrow) and delayed (**f**, **g**, black arrow) abnormal mineralization in homozygous mutant embryos

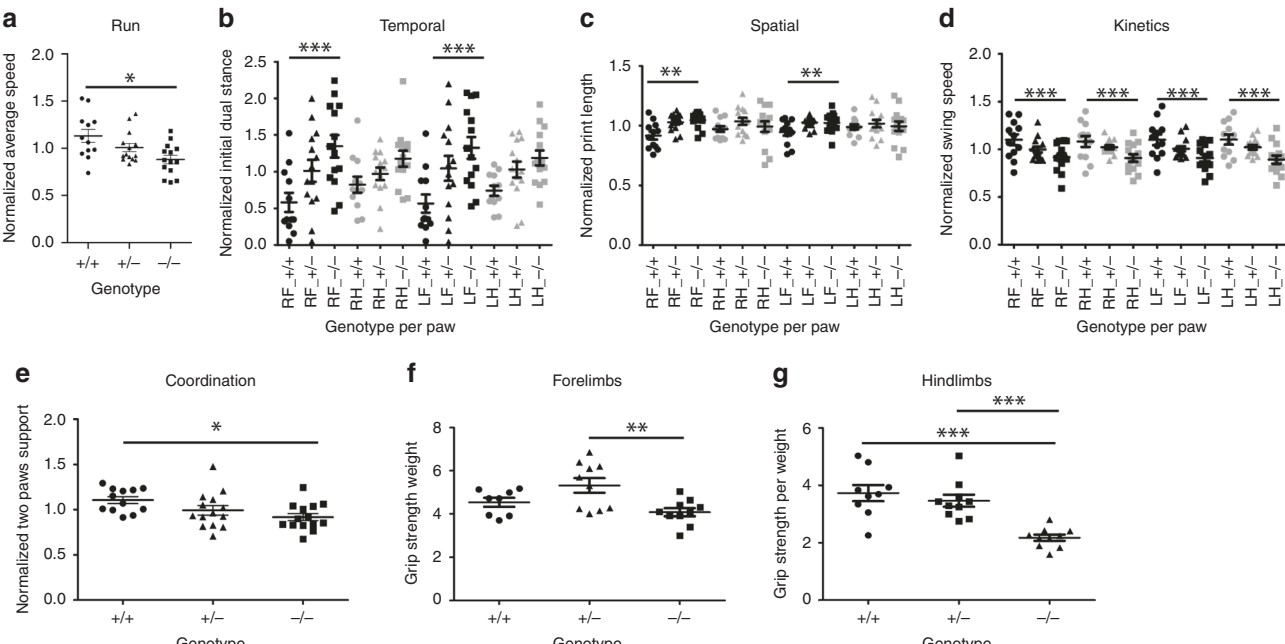

**Fig. 5** *Snap29* mutant female mice exhibit gait defects as measured by Catwalk. Catwalk assay was used to monitor gait parameters in $Snap29^{+/+}$, $Snap29^{+/-}$, and $Snap29^{-/-}$ females. The run characteristic parameter run average was significantly slower in $Snap29^{-/-}$ females (**a**). The temporal parameter initial dual stance was significantly elevated in both front paws (**b**). The spatial parameter print length was increased in both right and left front paws (**c**). The kinetic parameter swing speed was decreased in all paws of mutant animals (**d**). The interlimb coordination parameter support on two paws was significantly reduced in $Snap29^{-/-}$ females (**e**). The grip strength was assessed for fore limbs (**f**) and hind limbs (**g**). $Snap29^{-/-}$ females exhibited weaker fore and hind limbs than $Snap29^{+/+}$ animals. RF right front paw; RH right hind paw; LF left front paw, and LH left hind paw. RF and LF are in black and RH and LH are in gray. Statistical significance: *$p < 0.05$, **$p < 0.01$, and ***$p < 0.001$. Error bars represent standard error of the Mean

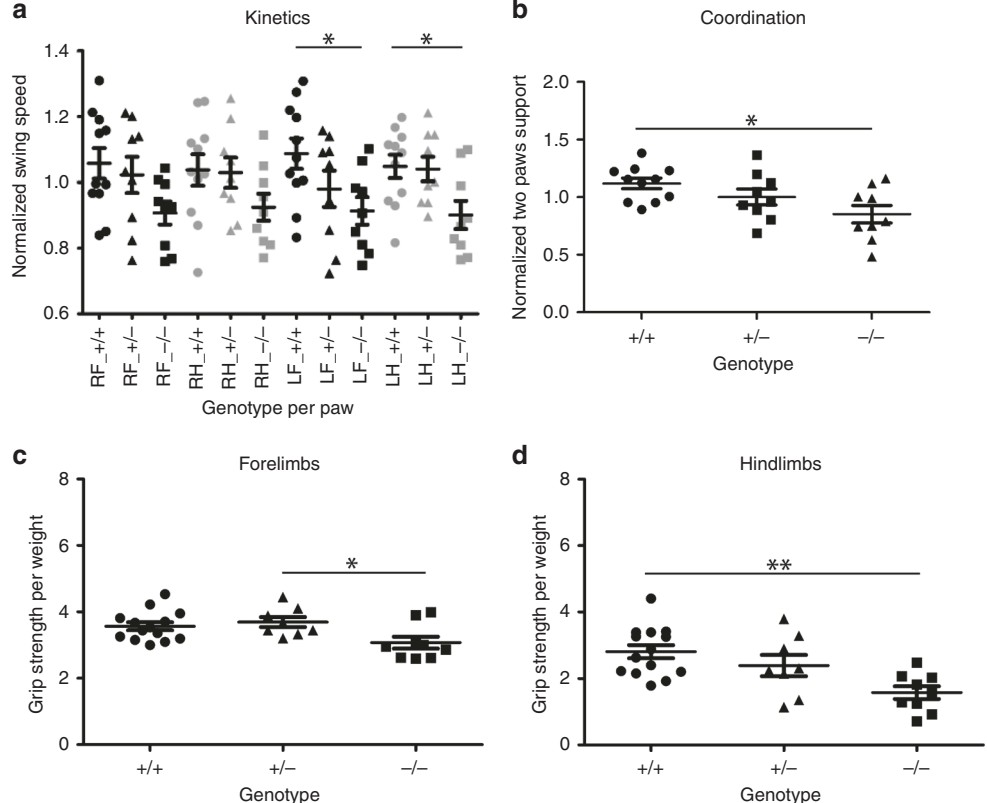

**Fig. 6** *Snap29* mutant male mice exhibit fewer gait defects as measured by Catwalk. Catwalk assay was used to monitor gait parameters in *Snap29*[+/+], *Snap29*[+/−], and *Snap29*[−/−] males. The kinetic parameter swing speed was decreased on the left side of *Snap29*[−/−] males (**a**). The interlimb coordination parameter support on two paws was significantly reduced in *Snap29*[−/−] males (**b**). The grip strength was assessed for fore limbs (**c**) and hind limbs (**d**). *Snap29*[−/−] males exhibited weaker fore and hind limbs than *Snap29*[+/+] animals. RF right front paw; RH right hind paw; LF left front paw; and LH left hind paw. RF and LF are in black and RH and LH are in gray. Statistical significance: *$p < 0.05$, **$p < 0.01$, and ***$p < 0.001$. Error bars represent Standard error of the Mean

Grip-strength was reduced in forelimb of *Snap29*[−/−] mutant males, ($p = 0.0217$; Fig. 6c) when compared with heterozygous litter mates ($n = 10$), while hindlimb strength was significantly reduced in homozygous mutant males (Fig. 6d; $p = 0.0025$) when they were compared with wildtype controls ($n = 9$). Similarly, grip strength was significantly reduced in forelimbs of female *Snap29*[−/−] mutant mice (Fig. 5f; $n = 10$; $p = 0.0068$) when compared with female heterozygous ($n = 10$), but not female wildtype controls ($n = 10$), while hindlimbs grip strength was significantly decreased in female *Snap29*[−/−] mutants when compared with female wild-type controls (Fig. 5g; $p = 0.0002$). We found that differences in hindlimbs grip strength persisted as animals aged. In fact, female *Snap29*[−/−] mutants retested at 14 week of age showed significantly decreased grip strength when compared with wild-type female (Supplementary Fig. 7E; $p < 0.0124$). In addition, female *Snap29*[−/−] mutants showed significantly reduced forelimbs grip strength when compared with female wild-type controls (Supplementary Fig. 7F; $p < 0.0291$). Altogether these data indicate that *Snap29* homozygous mutant mice present with reduced grip strength, first in their hindlimbs, and later in both hindlimbs and forelimbs as they age. Thus, we demonstrate that *Snap29*[−/−] mutants exhibit hypotonia, similar to CEDNIK patients.

**Retinal defects in *Snap29*[−/−] mutant mice.** As a subset of CEDNIK patients also show ophthalmological abnormalities such as optic nerve hypoplasia and atrophy, we used electroretinogram (ERG) to assess retinal function in 21 mice, including five wild type, eight *Snap29*[+/−] and eight *Snap29*[−/−] mutant mice. An attenuation

of the *b* wave in both scotopic and photopic ERG was observed in three out of eight (37.5%) of the homozygous mutant group. The average *b* wave amplitude of the three mice with abnormal ERG were lower than those of wild-type mice ($319.89 \pm 111.44\mu V$ in scotopic and $20.95 \pm 32\mu V$ in photopic vs. $628.97 \pm 113.45\,\mu V$ in scotopic and $107.82 \pm 35.6\mu V$ in photopic). However, the ERG results of the remaining homozygous and heterozygous mutant mice closely matched the results obtained from the wild-type group (Supplementary Table 4). Figure 7a–c shows representative scotopic and photopic ERG waveforms of wild type, heterozygous, and mutants (with low b amplitude). In addition, H&E revealed a significant thinning of the outer nuclear (ONL) and inner nuclear (INL) layers of the retina of *Snap29*[−/−] mutant mice compared with those of the wild-type group ($40.90 \pm 1.69\,\mu m$ vs. $46.15\,\mu m$, $p < 0.05$ in ONL and $30.87 \pm 1.26\,\mu m$ vs. $34.87\,\mu m$, $p < 0.05$ in INL) (Fig. 7g–i). Of the five *Snap29* homozygous mutant mice with normal ERG, three showed a thinning of the ONL and INL, one showed slightly reduced ONL and INL, and one showed normal ONL and INL. Thus, *Snap29* homozygous mutant mice model ophthalmological abnormalities found in CEDNIK although with reduced penetrance.

**Snap29[−/−] males are infertile.** During our studies, no live births were found when *Snap29*[−/−] mutant females and males were mated to each other. To determine if *Snap29*[−/−] females or males are infertile, we set up mating of *Snap29* homozygous mutant males and females with wild-type or heterozygous siblings for 3.9 months. No live births were found when mating pairs that consisted of a *Snap29*[−/−] mutant male ($n = 4$) regardless of the

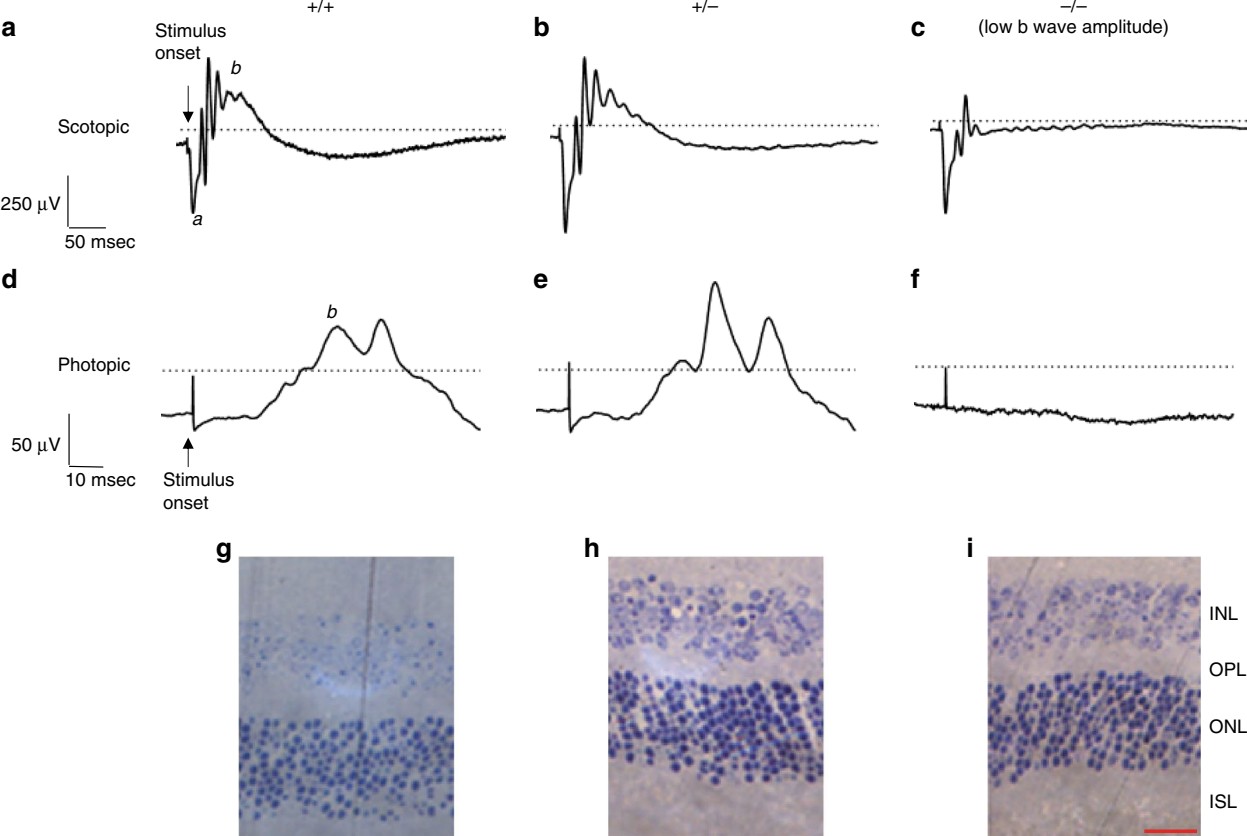

**Fig. 7** Analysis of ophthalmological defects in *Snap29*$^{-/-}$ mice. Representative scotopic (**a–c**; amplitude (in microvolt) and time (in millisecond) calibrations at the left of tracing a) and photopic (**d–f**; amplitude (in microvolt) and time (in millisecond) calibrations at the left of tracing d) ERG wave forms. **a** Scotopic ERG recorded from wild-type mouse; **b** Scotopic ERG recorded from heterozygous mouse; **c** Scotopic ERG recorded from homozygous mouse with low b wave amplitude; **d** photopic ERG recorded from wild-type mouse; **e** photopic ERG recorded from heterozygous mouse; **f** photopic ERG recorded from homozygous mouse with low b wave amplitude. In tracings a and d, the ERG a- and b-waves are identified with letters a and b, respectively. Representative eye histology of wild-type male mouse (**g**) at P118; heterozygous female at P83 (**h**) and homozygous female mouse at P87(**i**). ISL inner segment layer, ONL outer nuclear layer, OPL outer plexiform layer, INL inner nuclear layer. Scale bar represents 25 µm

genotype of the female: wild type ($n = 3$) or homozygous mutant ($n = 3$), indicating that *Snap29*$^{-/-}$ mutant males are infertile (Table 2). Importantly, mutant male mice were able to mount and generate vaginal plug in females, suggesting that weaker hindlimbs grip strength and/or lack of coordination was not the cause of their infertility.

Analysis of the reproductive organs of these males revealed that the testis/body ratio in *Snap29*$^{-/-}$ mutant mice were significantly less than those of *Snap29*$^{+/-}$ or wild-type males ($0.29 \times 10^{-2}$ vs $0.50 \times 10^{-2}$ and $0.54 \times 10^{-2}$, respectively; $p \leq 0.05$, Kruskal–Wallis ANOVA). In contrast, differences were not found in the weight of epididymis, seminal vesicles, coagulating glands, and prostate among the three genotypes (Supplementary Table 5). In addition, a subset of testis of *Snap29*$^{-/-}$ mutant mice revealed abnormal spermatogenesis with abnormal seminiferous tubules with degenerated germ cells, extensive vacuolization, loss of immature germ cells, and giant multinucleated cells (Fig. 8). Furthermore, the percentage of abnormal seminiferous tubules were higher in testis of *Snap29*$^{-/-}$ mutant males when compared with *Snap29*$^{+/-}$ or wild-type mice ($10.31 \pm 3.67$, $2.23 \pm 1.06$, and $0.15 \pm 0.10$, respectively; $p \leq 0.05$, Kruskal–Wallis ANOVA). In addition, the diameter of degenerated seminiferous tubules was reduced in *Snap29*$^{-/-}$ mutant (Fig. 8d) when compared with wild type (Fig. 8c), and few seminiferous tubules of testis of *Snap29*$^{-/-}$ mutant mice had spermatozoa in their lumen. Thus, SNAP29 is required for spermatogenesis and male fertility.

## Discussion

Mouse models are gold-star experimental models of human genetic conditions, and most abnormalities associated with a given human syndrome can be modeled when the orthologous gene is mutated in mice[13,14]. However, it is also clear that the genetic background of the mouse model can influence expressivity and penetrance of their abnormalities[15]. Thus, it is recommended that genetically engineered mouse models be analyzed on multiple genetic backgrounds, so as to identify any major roles played by differences in genetic backgrounds[15]. Before the current study, only one mouse model of a loss of function mutation in *Snap29* was reported and this model was made on the inbred C57Bl/6 genetic background. *Snap29* homozygous mutant mice on the C57Bl/6 genetic background exhibited ichthyosis and died at birth due to a failure in skin barrier formation[9]. Thus, on the C57Bl/6 genetic background neurological and motor defects that are associated with human mutations of *SNAP29* could not be modeled and characterized in adult *Snap29* homozygous mutant mice.

Herein, we report for the first time to our knowledge that *Snap29* is ubiquitously expressed during mouse development, which is surprising considering the specific defects associated with mutations in human. We also describe generation and characterization of *Snap29* mutant mice on a mixed CD1; FvB genetic background and show that abnormalities associated with mutations in *SNAP29* in patients with CEDNIK or 22q11.2DS are phenocopied in this *Snap29* mutant mouse line. Our

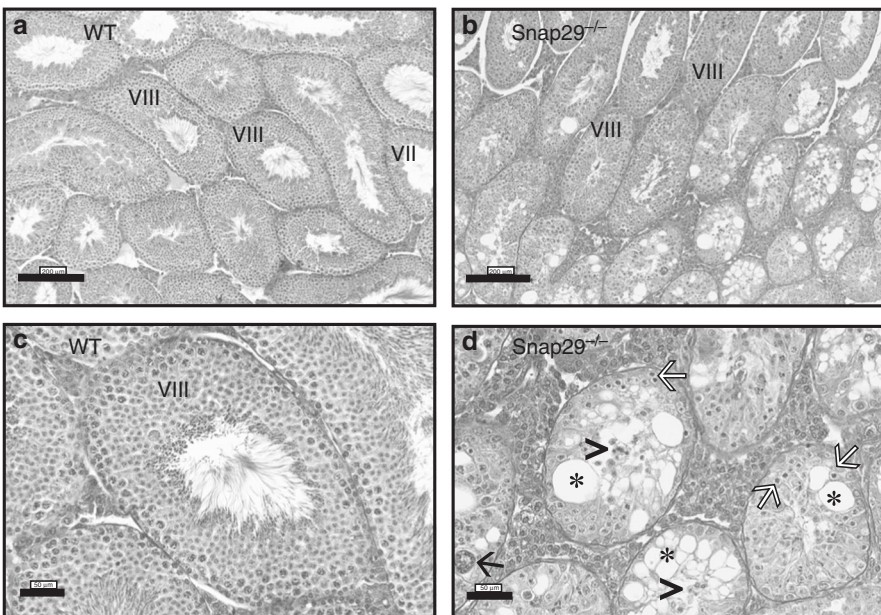

**Fig. 8** Histological analysis of *Snap29*$^{-/-}$ testis. Testis sections were stained with hematoxylin and eosin to evaluate spermatogenesis in WT (**a**, **c**) and *Snap29*$^{-/-}$ (**b**, **d**) mice. WT testis displayed normal spermatogenesis (Stages VII and VIII showing elongating spermatids and spermatozoa in the lumen). Seminiferous tubules in *Snap29*$^{-/-}$ testis had degenerated germ cells (white arrows), loss of immature germ cells accumulated in the lumen (arrow heads), giant multinucleated spermatids (black arrow) and extensive vacuolization (*). The diameter of degenerated seminiferous tubules was reduced in *Snap29*$^{-/-}$ (**d**) compared with WT (**c**) testis

**Table 2 Fertility of *Snap29* mutant mice**

| Mating # | Male genotype | Female genotype | Duration of mating | # litters |
|---|---|---|---|---|
| 1 | wt | wt | 5 months | 8 |
| 2 | wt | wt | 5 months | 8 |
| 3 | wt | wt | 2 months | 4 |
| 4 | wt | mut | 3 months | 6 |
| 5 | wt | mut | 4 months | 6 |
| 6 | htz | htz | 5 months | 13 |
| 7 | htz | htz | 5 months | 13 |
| 8 | htz | htz | 3 months | 9 |
| 9 | htz | htz | 3 months | 9 |
| 10 | htz | wt | 3 months | 0 |
| 11 | htz | mut | 3 months | 0 |
| 12 | mut | wt | 4 months | 0 |
| 13 | mut | mut | 4 months | 0 |
| 14 | mut | wt | 4 months | 0 |
| 15 | mut | mut | 4 months | 0 |
| 16 | mut | wt | 3 months | 0 |
| 17 | mut | mut | 3 months | 0 |
| 18 | mut | wt | 3 months | 0 |
| 19 | mut | mut | 3 months | 0 |

comprehensive analysis of *Snap29*$^{-/-}$ mutant mice reveals that many more abnormalities associated with CEDNIK syndrome are modeled in this mutant mouse line and clearly indicate that the genetic background does contribute to the expressivity and the penetrance of *Snap29* mutations in the mouse. Our analysis also reveals an unanticipated requirement for SNAP29 in the eye and the male reproductive tract.

Unlike *Snap29*$^{-/-}$ mutant on the inbred C57Bl/6 that all develops severe skin barrier defects and die neonatally[9], *Snap29*$^{-/-}$ mutants on the mixed CD1;FvB genetic background have a functional skin barrier at birth and the majority survives to adulthood. Importantly, on the C57Bl/6 genetic background,

expression levels of paralogs of SNAP29 are unaffected, suggesting that the loss of SNAP29 is not compensated by increased levels of its paralogs[16]. Since SNAP29 protein is not detectable in *Snap29*$^{-/-}$ mutant pups on both genetic background, we conclude that the presence of one or more modifier/s on the CD1; FvB genetic background plays a role in skin barrier formation. Thus, we postulate that neurological defects similar to those responsible for death of zebrafish *snap29* mutants[17], contribute to the death of *Snap29*$^{-/-}$ mutant pups which die within the first 7 days of life. However, as CEDNIK patients appear to die as infants primarily from complications arising from pneumonia[4], further studies of the cause of death of *Snap29*$^{-/-}$ mutant mice on the mixed CD1; FvB genetic background may shed insight into the underlying basis of death in CEDNIK patients.

We analyzed *Snap29*$^{-/-}$ mutant mice for motor defects associated with CEDNIK and 22q11.2DS. We demonstrated that adult *Snap29*$^{-/-}$ mutant mice displayed abnormal gait and hypotonia, similar to those found in CEDNIK patients. Furthermore, we found that *Snap29*$^{-/-}$ mutant male and female mice showed deficiencies in all five categories described by Gaballero-Garrido et al.[12]. Comparison of parameters with differences between *Snap29*$^{-/-}$ mutant mice and their wild-type litter mate revealed that those parameters with differences were shared with several mouse models of spinal cord injury and other neurological defects[18–20]. We did observe that some *Snap29*$^{-/-}$ mice sporadically suffered from seizure (Supplementary Movie 2). However, we did not detect any morphological changes during neurogenesis or in adult brains of *Snap29*$^{-/-}$ mutant mice (Supplementary Figs. 9–11). In fact, the only abnormalities found in the central nervous system were in the retina of eye. The absence of morphologically distinct defects in brains of *Snap29*$^{-/-}$ mutant mice could indicate that the role of *Snap29* in the cortex is not conserved in mice, or that neuronal abnormalities show very low penetrance on this genetic background. It is also possible that neurological defects found in *Snap29*$^{-/-}$ mutants are caused by impaired neurotransmitter exocytosis and abnormal synaptic transmission[21,22] or a more general

impairment of endocytic pathway, in which SNAP29 plays a crucial role[23]. Characterization of Snap29−/− mutant mice on other genetic backgrounds and/or mutation of Snap29 specifically in neurons will provide insights into the role of Snap29 in the central nervous system.

Unexpectedly, we found that all Snap29−/− mutant males were infertile. Thus, we postulate that SNAP29 may also be required during sperm maturation, since only 10% of seminiferous tubules of Snap29−/− mutant males were abnormal and this alone cannot explain the observed male infertility.

During vesicular transport, endocytosis, and exocytosis, SNAREs self-assemble into stable four-helix bundles between vesicular membranes and target membranes to mediate fusion of vesicle membranes to the target membrane. Trans-SNARE complex formation is also important for membrane fusion and completion of the acrosome reaction, a necessary event for fertilization[24]. Hence, we hypothesize that SNAP29 is required for spermatogenesis and molecular mechanisms associated with the acquisition of fertilizing ability by the spermatozoon. Future studies will focus on fully elucidating the participation of SNAP29 in spermatogenesis and sperm function.

We previously showed that hemizygous deletion of the 22q11.2 region uncovers pathogenic mutations in SNAP29, leading to CEDNIK phenotype in a subset of 22q11.2DS patients[7]. Intriguingly two patients with mutations in SNAP29 and hemizygous for 22q11.2 were atypical and did not have any skin abnormalities, a hallmark of CEDNIK, while only one patient had neurological abnormalities[7], suggesting that mutations of SNAP29 show incomplete penetrance. Although abnormalities and pathologies have not been reported in parents of CEDNIK patients whom are obligated carriers of SNAP29 mutations, the recent report of Sun et al.[8] that a heterozygous truncating mutation in SNAP29 was responsible for ADNFLE supports this hypothesis. Importantly, abnormalities were also found in Snap29+/− mice. In fact, heterozygotes pups had extra ribs and females showed interlimb coordination defects, indicating that this gene is also haploinsufficient in mice. Therefore, we propose that hemizygosity for SNAP29 in the majority of patients with 22q11.2DS contributes to a subset of abnormalities found in these patients. More specifically, our studies suggest that SNAP29 may play a role in seizures, eye and motor defects found in 22q11.2DS. As 90% of 22q11.2DS patients exhibit motor delays, we propose that our mouse model is a tremendous tool for investigating the etiology of motor defects in 22q11.2DS and strongly supports SNAP29 as a candidate gene for these deficits. Furthermore, it was recently reported that azoospermia may be associated with 22q11.2DS, and that male infertility may be an underappreciated phenotype associated with this syndrome[25], a phenotype that could be uncovered using our mice model. We propose that hemizygosity for SNAP29 maybe the underlying cause for male associated infertility in 22q11.2DS.

SNAP29 functions in a number of cellular pathways, including autophagosome-lysosomal fusion, endocytosis, and kinetochore formation[5,17,26,27]. Based on the work of Schiller et al., we postulate that abnormal SNAP29-regulated autophagy contributes to epidermal hypoplasia and disrupted lamellar body function[9], as was found in homozygous mutants on the inbred genetic background. However, it is unlikely that ER-stress induced autophagy alone is responsible for the multitude of abnormalities that we uncovered. Furthermore, though LC3B-II was increased in fibroblast of Snap29−/− mice[9], indicating an increase in autophagosomes, additional marker analysis suggested that this was most likely the result of ER-stress induced autophagy, and not a consequence of abnormal autophagosome-lysosome fusion. Thus, SNAP29 may be required at multiple steps in autophagy.

It is also possible that SNAP29-associated function in kinetochore formation[28], and endocytosis might contribute in a cell-type and tissue specific way to abnormalities found in Snap29−/− mutants. Future work will focus on identifying the timing and the specific cellular pathways disrupted in Snap29 mutant tissues.

## Methods

**Animals.** All procedures and experiments were performed according to the guidelines of the Canadian Council on Animal Care and approved by the Animal Care Committee of the RI-MUHC. CD1 mice were purchased from Charles River laboratories. Snap29 mutant mouse lines (Snap29lam/lam) were generated on a mixed genetic background (CD1; FvB) and maintained on the outbred CD1 genetic background.

**Generation of Snap29 in situ probe.** An in situ probe for Snap29 was generated from cDNA of wildtype E10.5 CD1 embryos. Briefly, RNA was extracted from embryos using Trizol (Invitrogen) according to the manufacturer's instructions. SuperScript® II Reverse Transcriptase (Thermo Fisher Scientific) Kit was used to synthesize a complementary DNA (cDNA). Primers were designed and used to amplify exons 2–5, including 207 bp of the 3′ UTR of Snap29. Primers used: forward sequence: AGCCCAACAGCAGATTGAAA and reverse sequence: AAAACTCAGCAGAACAGCTCAA. The cDNA fragment was cloned into TOPO using a TA Cloning Kit (Invitrogen). The cloned Snap29 cDNA was verified by Sanger sequencing. DIG RNA Labeling Mix (Roche) was used to produce digoxigenin labeled sense and antisense probes, according to the manufacturer's instructions. Briefly, the plasmid was linearized using either XbaI and Kpn1 restriction enzymes (NEB), and SP6 and T7 polymerases (NEB) were used to produce sense and antisense probes, respectively.

**In situ hybridization.** E7.5–E12.5 wild-type embryos were collected from pregnant CD1 females, fixed in 4% paraformaldehyde overnight at 4 °C and dehydrated in methanol (for whole mount) or ethanol (for in situ sections). For in situ hybridization on sections, deciduas and embryos were serially sectioned at 5 μm. Antisense probe was used to detect the expression of Snap29, and sense probe was used as a control. All protocols used for whole mount or section in situ hybridization were previously described[29].

**Generation of Snap29 knockout mice line using CRISPR/Cas9.** Two Snap29 mutant mouse lines (Snap29lam1 and Snap29lam2) were generated on a mixed genetic background (CD1;FvB) using CRISPR/Cas9[30]. Briefly, four single guide RNA (sgRNA) sequences flanking exon 2 of the mouse Snap29 gene, (sgRNA1 and sgRNA2 located in intron 1, and sgRNA3 and sgRNA4 located in intron 2) were designed using the online services of Massachusetts Institute of Technology (http://crispr.mit.edu). SgRNAs were synthesized using GeneArt Precision gRNA Synthesis Kit (Thermo Fisher Scientific). 50 ng/ul Cas9 mRNA (Sigma) together with 4 sgRNAs (6.25 ng/μl) were microinjected into the pronucleus of fertilized eggs collected from mating of wild-type CD1 and FvB mice (Fig. 1b). Injected embryos were transferred into uterus of pseudo-pregnant foster mothers (CD1, Charles River laboratories). Sanger-sequencing revealed deletions of 464 and 517 bp, in two mutant mouse lines Snap29lam1 and Snap29lam2. Since our analysis of Snap29lam1 and Snap29lam2 mutant embryos and pups revealed similar penetrance and expressivity in phenotypes after six-generation of backcrossing to CD1, these numbers have been combined. All mutant mouse lines were maintained on the mixed CD1 genetic background.

**Genotyping.** Yolk sacs and tail tips were used for genotyping embryos and pups from the Snap29 colony, respectively. Standard polymerase chain reaction (PCR) was used for genotyping. Three-primers PCR (primers sequences: Right primer: GACTGAGTCTCACCTGGTCC, Left primer: TGGCTTTTGGAATGACTTG, was optimized to enable detection of embryos and pups carrying wild-type (435 bp amplicon) or mutant alleles (240 and 300 bp amplicons, Snap29lam1 and Snap29lam2, respectively) (Supplementary Fig, 11). All PCR products were confirmed by Sanger Sequencing.

**Embryo, brain and testis collection.** Wild-type CD1 embryos were used for all in situ hybridization experiments. For all other analysis, embryos were collected from Snap29 mutants on a mixed genetic background (CD1; FvB). For embryo collection, the day of plug was used to indicate pregnancy and designated as embryonic day (E) 0.5. Homozygous mutant embryos and pups were generated from mating of Snap29 heterozygous mice. For perinatal and adult analysis, the day of birth was designated as postnatal day (P) 0. Embryos and tissues (brain and skin) were collected in RIPA for western blot analysis or fixed in 4% paraformaldehyde and processed as previously described[29]. For testis collection wild-type or Snap29 homozygous mutant male mice were euthanized and testes were dissected, weighed, and fixed immediately with Bouin fixative for 24 h before processing and embedding in paraffin blocks as previously described[31].

**Western blot analysis**. Western blot analysis was performed as previously described. Briefly, P1 dorsal skin were collected in $1 \times$ RIPA lysis buffer (25 mM Tris-HCl pH7.6, 10% glycerol, 420 mM NaCl, 2 mM $MgCl_2$, 0.5% NP-40, 0.5% Triton X-100, 1 mM EDTA, protease inhibitor) on ice. Skin samples were then sonicated and centrifuged at 13000 rpm for 20 min at 4 °C. Clarified protein lysates were measured according to standard methods using a DC protein assay kit (Bio-rad, Mississauga, Ontario, Canada). Bradford assays (Bio-Rad) were used to standardize amount of total protein loaded on a gel 12% polyacrylamide gel was then transferred to polyvenyidene fluoride (PVDF membrane, Bio-Rad). Five percent nonfat dry milk in PBST (Phosphate buffered saline with Tween 20) was used to block membranes followed by incubation with primary and secondary antibodies. The ECL Western Blotting Detection System (ZmTech scientifique) was utilized to detect the immunoreactive bands and images were taken with Bio-Rad's ChemiDoc MP System (catalog# 1708280). Primary antibodies used for western blots were rabbit anti-SNAP29 monoclonal antibody (1:5000, Abcam #138500) and rabbit anti-Beta-actin monoclonal antibody (1:5000, Cell Signaling #4970). An anti-rabbit secondary antibody (1:5000, Cell Signaling #7074) was used for both SNAP29 and beta-actin. The bands were quantified by Image lab (Bio-Rad). Statistical significance was tested using Prism 5 (GraphPadVersion 1.0.0.0).

**Transmission Electron Microscopy (TEM)**. P1 pups were anesthetized and perfused with 2.5% glutaraldehyde in 0.1 M sodium cacodylate buffer. 1-mm$^3$ dorsal skin section was taken from the perfused pups and placed in the same fixative for an additional 2 h at 4 °C. The skin samples were rinsed three times in 0.2 M sucrose/ sodium cacodylate buffer and were left in this buffer overnight. The samples were then post-fixed in ferrocyanide-reduced osmium tetroxide for 1 h at 4 °C followed by dehydration with acetone, infiltration with acetone/epon, and embedded in Epon. Sections were cut with Leica Microsystems Ultracut UCT. To examine the sections, FEI Tecnai 12 BioTwin 120 kV Transmission Electron Microscope imaged with an AMT XR80C CCD Camera System.

**X-Gal epidermal barrier assay**. Freshly collected E17.5 embryos and P1 pups were weighed and washed with PBS before performing the skin permeabilization assay[32]. Briefly, embryos or pups were incubated overnight in X-Gal staining solution (1 mg/ml of X-Gal, 4 mM $K_4Fe(CN)_6.3H_2O$, 4 mM $K_3Fe(CN)_6$, 2 mM $MgCl_2$ in PBS pH4.5) at 37 °C. After incubation the embryos and pups were washed in PBS 1 to 2 min and assessed for blue colored stain and were photographed with an Infinity1 camera mounted on a LEICA MZ6.

**Skeletal preparations**. Skeletal preparations with Alcian blue to stain cartilage or Alcian blue/Alizarin red to stain cartilages and skeletons of E14.5, E16.5, P1 and P3 embryo and pups were performed as previously described[33]. Briefly, whole mount E14.5, E16.5 embryos and P1 and P3 pups were treated with ethanol and acetone for 24 h after eviscerating and removal of skin (for E16.5, P1 and P3 samples) and stained with Alcian Blue/Alizarin Red stain (Sigma) for 3–4 days at 37 °C on a rocker. Stained embryos and pups were incubated in 1% KOH for 72–96 h and washed with 1% KOH/glycerol mixture.

**Rotarod**. Rotarod (47600, KYS Technology, UGO Basile S.R.L, Italy) testing was performed as previously described[34]. Briefly, mice were tested on an accelerating rotarod mode (4–40 rpm) for 300 sec. The latency to fall (in seconds) was recorded. Two rotations in the cylinder were considered as a fall. Each mouse had 3 days training with 3 trials per day and the fourth day was the experimental day. The mean value of the three trials on the experimental day was used for statistical analysis.

**Grips strength measurement**. Mice were held by the base of their tail and lowered toward the mesh of the grip strength meter (Bioseb Model GS3). After grasping with their forepaws, the body of the mouse was lowered to be at a 45-degree position with the mesh. The mouse was then pulled by the tail away from the mesh until the grip was broken. Similarly, after grasping with the hind paws, the mice were lowered at the 45-degree position with the bar and pulled until the grip was broken. Nine trials were performed for each mouse and the average was used as the grip strength score for that mouse[28].

**Immunohistochemistry and histological staining of brain and eye**. Paraffin sections (E17.5, coronal, 6 μm) were used for immunohistochemistry as previously described with slight modification[35]. Briefly, after deparaffinization and rehydration, the sections were incubated with 1% $H_2O_2$/methanol for 10 min Antigen retrieval was performed by heating sections at 100ºC in 0.01 M citric acid for 10 min Sections were blocked with 10% goat serum for 1 h, incubated with primary antibodies at 4ºC overnight, then incubated with appropriate secondary antibody conjugated with biotin (1:1000, Vector Lab) at RT for 1 h, and incubated with ABC kit (1:1000, Vector Laboratories, PK-6100) at RT for 1 h. For nuclear staining, sections were treated with 0.5% Triton X-100 for 10 min before the blocking step. The signal was detected with immPACT DAB (Vector Laboratories, SK-4105) and each section was counterstained with diluted Hematoxylin (1:30, Leica Biosystems, 3801570). The primary antibodies used in this study are TBR1 (rabbit, 1:1000,

Abcam, ab31940), CTIP2 (rat, 1:200, Abcam, ab18465), Reelin (mouse, 1:200, Millipore, MAB5364), and SATB2 (mouse, 1:200, Bio Matrix Research, BMR00263).

Coronal sections (5 μm) of E17.5 wild-type and homozygous *Snap29* mouse embryo brains were stained with cresyl violet in histopathology core of MUHC (McGill University Health Center).

For eye histological section, mice were sacrificed following the ERG recording, and the eyes were removed and fixed for 2 h in 4% paraformaldehyde. After a dissection to remove the cornea and lens, the eye cups were placed in 4% paraformaldehyde and fixed overnight on a shaker. On the following day, the eyecups were dyed with 1% osmium tetroxide for 3 h and subsequently dehydrated with 50, 80, 90, 95 and 100% ethanol. Finally, the eye cups were embedded in resin (Durcupan ACM Fluka epoxy resin kit, Sigma-Aldrich, Canada) and placed in an oven at 55 °C for 48 h. The resin embedded eye blocks were then cut into 1.0 μm thick sections with an ultramicrotome (Leica EM UC6 microtome, Leica microsystem, USA) and dyed with 0.1% toluidine blue. Retinal photographs were taken using a Zeiss Axiophot (Zeiss microscope, Germany) at a 40X magnification. The thickness of each retinal layer was measured in the supero-temporal region at a position between 680 μm and 1020 μm from the optic nerve head[36].

**Electroretinogram (ERG) recordings**. After an overnight dark adaptation, flash ERGs were recorded as previously reported[36]. Briefly, the mice were anesthetized with an intramuscular injection of ketamine (75 mg /kg) and xylazine (12.5 mg/kg) solution. One drop of 1% Mydiacyl and 0.5% Alcaine were used respectively to dilate the pupil and to decrease blinking. The mice were then placed on their right side on a homeothermic heating pad (Harvard Apparatus, Holliston, MA) at a fixed temperature of 37 °C in a recording chamber of our design. ERGs were recorded by placing a DTL electrode fiber (27/7 X-Static® silver-coated conductive nylon thread, Sauquoit Industries, Scranton, PA, USA) on the cornea and maintained in position with a coat of moisturizing gel (Tear-Gel Novartis Ophthalmic, Novartis Pharmaceuticals Inc., Canada). A reference electrode (Grass E5 disc electrode) was placed under the tongue of the mouse and a ground electrode (Grass E2 subdermal electrode) was inserted subcutaneously in the tail. Full-field ERGs (bandwidth: 1–1000 Hz, 10,000X, 6db attenuation, Grass P-511 amplifiers) were recorded using the Biopac Data Acquisition System (Biopac MP 100WS, Biopac System Inc., Goleta, CA, USA). Scotopic ERGs were evoked to gradually brighter flashes of white light ranging from $-6.3$ log cds.m$^{-2}$ to 0.9 log cds.m$^{-2}$ in 0.3 log-unit intervals (Grass PS-22 photo-stimulator, Grass Technologies, Warwick, RI, USA; average of 5 flashes, interval of stimuli: 10 sec). Photopic ERGs (background light: 30 cd.m$^{-2}$ flash intensity: 0.9 cds.m$^{-2}$; average of 20 flashes at 1 flash per second) were recorded after a light adaptation period of 20 min following the dark-adaptation period. The amplitude of the a-wave was measured from the level of the baseline to the most negative trough while the amplitude of the b-wave is measured from the trough of a-wave to the fourth positive peak.

**Statistics and reproducibility**. To assess statistical significance of the different parameters used in this study, we first determined if the data for a given parameter was normally distributed using the D'Agostino and Pearson normality test. We performed ANOVA analysis on normally distributed data followed by the Tukey post-hoc analysis to determine which pair of data was statistically different. If the data did not pass the normality test, we performed a Kruskal–Wallis test followed by a Dunn's post hoc test to determine which pair of data was statistically different. The $p$-values of the statistically significant data are represented by *; * $\leq 0.01$, ** $\leq 0.001$ and *** $\leq 0.0001$.

Unless, otherwise noted data for experiments are from a minimum of three litters, from three distinct mating pairs. We found that data were reproducible within litters and generations.

**Reporting summary**. Further information on research design is available in the Nature Research Reporting Summary linked to this article.

## Data availability
All relevant data generated or analyzed during this study are included in this article. More supporting data are available upon reasonable request from the corresponding author. Source data underlying plots shown in Figs. 1, 5, and 6 are provided in Supplementary Data 1.

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

## Acknowledgements

We would like to thank Dr Mitra Cowan, Associate Director of the transgenic core facility at McGill University, Goodman Cancer Center, for the micro-injection experiments. V.K. was supported by the Government of Azerbaijan. This work was supported by a grant from the Canadian Institutes of Health Research (MOP#142452). We would like to thank Dr Sebire for help in brain collection, Ildi Troka in the lab of Dr Hendy for help with grip-strength, and Dr Braverman for use of the Rotarod. This work would not be possible without help by Mathieu Simard of the McGill Small Animal Imaging Lab. We thank Drs McDonald-McGinn and Majewski for support and Drs Slim, Rosenblatt, and Beauchamp for reading of the manuscript. L.J.M. is a member of the Research Centre of the McGill University Health Centre which is supported in part by FQRS.

## Author contributions

L.J.M conceived the study. V.K., J.F.B.L., S.S.A., Y.L., E.S., A.K., Z. T. F. and S.L. performed experiments and analyzed data. P.L., C.O., J.A.G. and L.J.M. supervised the study and help analyzed data. L.J.M. wrote the paper with contributions from all the authors.

## Competing interests

The authors declare no competing interests.
