## [Peer Review File · Communications Biology]

Reviewers' comments:

Reviewer #1 (Remarks to the Author):

In their paper, Keser et al report the generation of Snap29 mutant mice. They show that contrary to previous evidence these survive to adulthood and present a range of phenotypes that recapitulate the trait of CEDNIK and other congenital syndromes characterized by loss or diminished Snap29 activity. In addition, they describe a male sterility as a novel phenotype.

While Snap29 KO mice have been previously reported, their analysis has been limited to skin phenotypes or early lethality has prevented full characterization of phenotypes that might clue in on the pathogenesis of CEDNIK. In such context, Keser et al provide an additional, valid model. Their study is limited to a phenotypic characterization. However, it has the merit of:

- Beginning to address the trait variability observed in patients;
- link Snap29 to spermatogenesis
- starting to explore the pathogenesis of defects other than the skin ones.

The experiments are accurate and well presented, but the paper somehow misses the chance to capitalize over some of the interesting insights presented (see below).

Points to address:

while previous models showed neonatal lethality, a portion of mutant mice here survive to adulthood and can mate, something never reported in the case of CEDNIK patients. Skin barrier and neuromotor defects are variable. It is not clear why this is the case, but sure for future therapies it would be great to know more about the origin of such variability.. Could the authors check whether paralogs of Snap29 are overexpressed as compensation relative to control? This experiment would require few q-pcr and western blot and might get a clue on strategies to ameliorate the trait of CEDNIK patients. No cell biological function of Snap29 is tested for defects. Do cells accumulate undigested autophagosomes? Is there an increased amount of apoptosis during development or post natal? These questions could be addressed with few cell bio assays and would go a long way towards understanding how Snap29 function could be compensated.

The novel function in spermatogenesis suggest that a trafficking function of Snap29 might support sperm development and perhaps acrosome formation as speculated. In analogy to what they have done with skin sections,. the authors should use EM to gain sufficient resolution to see whether acrosomes are formed and what types of organelles are misdeposited in mutant testis cells Phenotypes possibly attributed to hets should be better discussed as they have not been previously described

Others in different models have reported ubiquitous expression of Snap29 and it is not surprising because as far as we know at least for autophagy there no other QbQc SNARE described to enable fusion of autophagosomes to lysosomes a process occurring in most cells. Authors should rather comment on why some of lack of functions of Snap29 become critical to certain processes in certain cells during development and postnatal life.

Reviewer #2 (Remarks to the Author):

This manuscript describes the generation of an excellent mouse model of human disorder related to loss of the Snap29 gene called CEDNIK (cerebral dysgenesis, neuropathy, ichthyosis, and keratoderma). In both mouse mutants and human patients, there are neurological, ophthalmological, and skin disorders. Other investigators had created a Snap29 mutant line on an inbred C57Bl6 background that die at birth so only the skin phenotype was analyzed. Here the authors created a Snap29 exon deletion mutant and used a genetic background that results in viability past birth to allow characterization of the role of Snap29 in multiple organs and tissues in juvenile and adult animals.

The authors describe the overt phenotype of Snap29 mouse mutants and using SEM, histology, and functional/behavioral studies to evaluate the skin, eye, testes, brain and motor function. They appropriately assess male and female mice and compare wildtype, het and homozygous mutants. This descriptive paper clearly shows that the Snap29 mice recapitulate many of the human phenotypes and hence will be a valuable model to understand deficits in human patients. Moreover, the authors document in a rigorous way that the various organ defects can show incomplete penetrance and

expressivity, similar to that observed in humans.

Minor comments:

Add the catalog number for the antibodies used

Line 374: liter should be litter

Line 410-412: is a thinning of the ONL and INL observed in homozygotes that showed a normal b wave amplitude or only in animals with an abnormal b wave amplitude?

Line 471 zebrafish model requires a reference citation

Tables 2 & 3: please indicate the age of analysis, as multiple ages were reported in the text

Table 4 legend: define RH, LH, etc.

Reviewer

Lee Niswander

Dear Dr. Jung-Eun,

First, we would like to thank you and the reviewers for the very quick review and helpful comments on our manuscript. Below we have addressed the reviewers' comments to the best of our abilities.

Reviewer #1 (Remarks to the Author):

The experiments are accurate and well presented, but the paper somehow misses the chance to capitalize over some of the interesting insights presented (see below).

Points to address:

“while previous models showed neonatal lethality, a portion of mutant mice here survive to adulthood and can mate, something never reported in the case of CEDNIK patients. Skin barrier and neuromotor defects are variable. It is not clear why this is the case, but sure for future therapies it would be great to know more about the origin of such variability”

We agree with the reviewer that this is an interesting observation, however, we feel that any discussion on why this variability is found would be extremely speculative. In the future we think that this new model could be used for a number of genetic experiments to try and identify alleles on this mixed genetic background that contribute to the genetic variability that we observed.

Could the authors check whether paralogs of Snap29 are overexpressed as compensation relative to control? This experiments would require few q-pcr and western blot and might get a clue on strategies to ameliorate the trait of CEDNIK patients.

We agree that overexpression of SNAP29 paralogs as a compensatory mechanism for the loss of SNAP29, is an obvious and attractive hypothesis. However, this is unlikely to be the case. In fact, Nandhini Sivakumar “PhD thesis title -Synergistic SNARE modulators of Neurotransmission, 2015”, tested this hypothesis in two different mutant mouse models in her thesis. Nandhini Sivakumar showed that deletion of exon 2 of *Snap29* in the hippocampus and cerebral cortex lead to a significant reduction of SNAP29 protein, but no compensatory changes in protein levels of SNAP23, 25, and 47 (Figure 12). Similarly, no significant difference was found in expression of these proteins in cortical lysates from constitutive *Snap29* homozygous mutant mice on the C57Bl/6 genetic background (Figure 16). Since expression of SNAP29 was reduced or completely abrogated in both models (similar to our model) we predict that loss of *Snap29* does not perturb expression of the paralog genes. Although we could repeat the experiments to confirm, the price of the antibodies for SNAP23, 25 and 47 make this experimentation cost prohibitive and would not change our conclusion: the lack of SNAP29 function that is responsible for the phenotypes found.

No cell biological function of Snap29 is tested for defects. Do cells accumulate undigested autophagosomes? Is there an increased amount of apoptosis during development or post natal?

These questions could be addressed with few cell bio assays and would go a long way towards understanding how Snap29 function could be compensated.

We agree with the reviewer that these are important mechanisms which need to be studied. However, this needs to be addressed in every tissue with a phenotype. The multiple roles described for SNAP29 suggest that it may have tissue-specific function. However, it is worth noting that for skin defects, undigested autophagosomes was not found although an increase in ER stress was reported. Given that we observed similar skin phenotypes by TEM, we believe that would also be the case in epidermal cells in *Snap29* homozygous mutants a mixed genetic background. Furthermore, we showed that there were no significant changes in morphology or proliferation in the cortex of homozygous mutant mice, consistent with the PhD thesis work of Nandhini Sivakumar. Therefore, we believe that pursuing the biological function of *Snap29* in all tissues with abnormalities is beyond the scope of this paper.

The novel function in spermatogenesis suggest that a trafficking function of Snap29 might support sperm development and perhaps acrosome formation as speculated. In analogy to what they have done with skin sections. the authors should use EM to gain sufficient resolution to see whether acrosomes are formed and what types of organelles are misdeposited in mutant testis cells.

We are also very excited about this phenotype and have experiments underway to address this point. This will be the subject of a follow-up manuscript.

Phenotypes possibly attributed to hets should be better discussed as they have not been previously described. **We agree with the reviewer and have made modifications to the text (in the discussion section) we hope that the have address this point.**

Others in different models have reported ubiquitous expression of Snap29 and it is not surprising because as far as we know at least for autophagy there no other QbQc SNARE described to enable fusion of autophagosomes to lysosomes a process occurring in most cells. Authors should rather comment on why some of lack of functions of Snap29 become critical to certain processes in certain cells during development and postnatal life. **We have added a section to the discussion to address this point.**

Reviewer #2 (Remarks to the Author):

Minor comments:

We have made all of the modifications recommended by Dr. Niswander.

Add the catalog number for the antibodies used

This information has been added.

Line 374: liter should be litter

This typo has been fixed.

Line 410-412: is a thinning of the ONL and INL observed in homozygotes that showed a normal b wave amplitude or only in animals with an abnormal b wave amplitude?

We have added this sentence to clarify the statement at line 419: “Of the 5 *Snap29* homozygous mutant mice with normal ERG, 3 showed a thinning of the ONL and INL, one showed slightly reduced ONL and INL, and one showed normal ONL and INL.”

Line 471 zebrafish model requires a reference citation

We have added the missing citation.

Tables 2 & 3: please indicate the age of analysis, as multiple ages were reported in the text

We have added the age of the pups and mice analyzed.

Table 4 legend: define RH, LH, etc.

We have defined all abbreviations in the figure legend.

REVIEWERS' COMMENTS:

Reviewer #1 (Remarks to the Author):

The authors address my points in their cover letter.

However:

1. about my questions regarding redundancy with paralogs, they quote unpublished work present in a 2015 phd thesis that I managed to download (not provided). The thesis indeed address my concerns in a similar, albeit not identical, genetic background. If they want to lean on these data at the very least they should modify the discussion to address the point a reference the thesis.

2. about my questions regarding followup on the acrosome function (the only novel data in addition to the not explained viability of their mice), they let me know that that will be part of a followup manuscript.

Text modifications have been done but no additional experimental work has been added.

Reviewer #2 (Remarks to the Author):

my concerns, which were minor, have been satisfactorily addressed.

To address the reviewer #1's comments, "I ask you to please cite the 2015 phd thesis and..."

To address the comment we have include this sentence and added the ph.d thesis reference at line 273:

"Importantly, on the C57Bl/6 genetic background, expression levels of paralogs of SNAP29 are unaffected, suggesting that the loss of SNAP29 is not compensated by increased levels of its paralogs¹⁶."

to include acrosome data to the final version of the manuscript if they are available from your collaborator.

To address the second comment: Unfortunately, acrosome data is not available at this time from our collaborator.

Reviewer #1 (Remarks to the Author):

The authors address my points in their cover letter.

However:

1. about my questions regarding redundancy with paralogs, they quote unpublished work present in a 2015 phd thesis that I managed to download (not provided). The thesis indeed address my concerns in a similar, albeit not identical, genetic background. If they want to lean on these data at the very least they should modify the discussion to address the point a reference the thesis.

As stated above, the reference and a sentence were added at line 273 of the manuscript in the discussion section.

2. about my questions regarding followup on the acrosome function (the only novel data in addition to the not explained viability of their mice), they let me know that that will be part of a followup manuscript.

Although we agree that fertility is a novel finding, we strongly disagree that we are presenting only 2 novel data on *Snap29* function. We have shown that *Snap29* mutant mice have motor defects. These motors defects are not accompanied by gross morphological defects unlike human patients. Furthermore, we report that *Snap29* mutant mice have ophthalmological problems. We also show that the reported lethality of *Snap29* mutant mice is likely not due to skin permeability and desiccation. Finally, we provided evidence that *Snap29* mutant mice are infertile. The mechanism behind this infertility will be addressed in a follow up paper from our collaborator.